# Diarrhoea among Children Aged under Five Years and Risk Factors in Informal Settlements: A Cross-Sectional Study in Cape Town, South Africa

**DOI:** 10.3390/ijerph18116043

**Published:** 2021-06-04

**Authors:** Thi Yen Chi Nguyen, Bamidele Oladapo Fagbayigbo, Guéladio Cissé, Nesre Redi, Samuel Fuhrimann, John Okedi, Christian Schindler, Martin Röösli, Neil Philip Armitage, Kirsty Carden, Mohamed Aqiel Dalvie

**Affiliations:** 1Department of Epidemiology and Public Health, Swiss Tropical and Public Health Institute, CH-4002 Basel, Switzerland; redin@who.int (N.R.); samuel.fuhrimann@swisstph.ch (S.F.); christian.schindler@swisstph.ch (C.S.); martin.roosli@swisstph.ch (M.R.); 2University of Basel, CH-4003 Basel, Switzerland; 3Centre for Environmental and Occupational Health Research, School of Public Health and Family Medicine, University of Cape Town, Cape Town 7925, South Africa; bamidele.fagbayigbo@uct.ac.za (B.O.F.); aqiel.dalvie@uct.ac.za (M.A.D.); 4Future Water Institute, University of Cape Town, Cape Town 7700, South Africa; john.okedi@uct.ac.za (J.O.); neil.armitage@uct.ac.za (N.P.A.); kirsty.carden@uct.ac.za (K.C.)

**Keywords:** children, diarrhoea, formal settlement, hygiene, informal settlement, sanitation, water quality

## Abstract

Background: There is limited data on the association between diarrhoea among children aged under five years (U5D) and water use, sanitation, hygiene, and socio-economics factors in low-income communities. The study investigated U5D and the associated risk factors in the Zeekoe catchment in Cape Town, South Africa. Methods: A cross-sectional study was conducted in 707 households in six informal settlements (IS) two formal settlements (FS) (March–June 2017). Results: Most IS households used public taps (74.4%) and shared toilets (93.0%), while FS households used piped water on premises (89.6%) and private toilets (98.3%). IS respondents had higher average hand-washing scores than those of FS (0.04 vs. −0.14, *p* = 0.02). The overall U5D prevalence was 15.3% (range: 8.6%–24.2%) and was higher in FS than in IS (21.2% vs. 13.4%, respectively, *p* = 0.01). Water storage >12 h was associated with increasing U5D (OR = 1.88, 95% CI 1.00–3.55, *p* = 0.05). Water treatment (OR = 0.57, 95%CI 0.34–0.97, *p* = 0.04), good hand-washing practices (OR = 0.59, 95%CI 0.42–0.82, *p* = 0.002) and Hepatitis A vaccination (OR = 0.51, 95%CI 0.28–0.9, *p* = 0.02) had significant preventing effects on U5D. Conclusions: The study highlights that good hygiene practice is a key intervention against U5D in informal settlements. The promotion of hand-washing, proper water storage, and hygienic breastfeeding is highly recommended.

## 1. Introduction

In 2017, diarrhoea was among the leading cause of death among all ages globally (more than 1.7 million deaths) and the fifth leading cause of death among children younger than five years with more than 0.44 million deaths) [1]. According to the World Health Organization (WHO) 2017 report, more than a quarter (26.9%) of diarrhoea deaths occurred among children younger than five years, and about 90% of diarrhoea deaths occurred in South Asia and sub-Saharan Africa [2]. Statistics South Africa (Stats SA) estimates that 20% of under-five deaths may be attributed to diarrhoea [3], with 31,436 diarrhoea cases recorded in 2016 [4], while other sources estimate the mortality rate between 8%–24% [5,6].

There is an unequal geographical distribution of diarrhoea burden among children younger than five years of age across Africa [6]. Children in low- and middle-income countries (LMICs) are the most vulnerable to diarrhoea. More than half of all the diarrhoea-related deaths among children in Africa were estimated to occur in 7.0% of the first-level administrative subdivisions (i.e., states, regions or provinces depending on the country) on the continent [6]. Diarrhoea is closely associated with environmental and socio-economic conditions, with the impoverished communities being most affected [7]. In 2016, over 75% (101/133) of diseases or disease groups listed in the Global Health Observatory had significant links with the environment, and more than one-quarter of the 6.6 million under-five child deaths were associated with environment-related causes and conditions [8]. Among diseases contributing to the environmental burden of diseases, diarrhoea diseases count for 22% and parasitic and vector-borne diseases for 12% [9].

Informal Settlements (IS) in Africa are synonymous with public health challenges as a result of dense population and poor sanitation [10,11,12]. The urban population growth rates in Africa are the highest in the world, and it has been projected that by 2050, Africa’s cities will be home to an estimated 950 million people [13]. In South Africa, 3.5 million out of the 17 million households (20%) were living in rural and informal settlements. In 2013, the Cape Town metro alone had 204 informal settlements with more than 200,000 households estimated to be living in conditions where fire and sanitation are among major health hazards, and this number increases each year [14].

In 2016, the population of Cape Town was estimated at 4,617,560 inhabitants with a 2.40% annual growth rate, in which 17% of the population was located in informal dwellings [12]. Many of these informal settlements are located on low-lying, flood-prone areas or steep slopes with poor water and sanitation services. Generally, those informal settlements are characterized by dense populations, unhygienic conditions, improper disposal of waste, and poor sanitation. With such living conditions, citizens who live in those areas are particularly vulnerable to diarrhoea. Additionally, a lack of access to potable water supplies, contamination risks associated with the water sources, poor sanitation, poor drainage, presence of contaminated grey water, and inadequate stormwater management, among other factors, may increase the risks for diarrhea—especially for children under five. During 2012–2015, there were an estimated 25,000–30,000 diarrhoea cases each year in children aged less than five years [15]. In 2017, Nerse et al. found the one and two-week prevalence of diarrhea among under-five children in informal settlements surrounding Cape Town were 13.22% and 16.9%, respectively [16].

Diarrhoea is a major public health issue in the Western Cape, South Africa, with a seasonal variation that has a peak between March and June every year. However, the sources and burden of early childhood diarrhoea are multidirectional and difficult to truly ascertain [17,18], and further, less is known about the situation in informal settlements. In this study, we set out to assess under-five childhood diarrhoea and its risk factors in the informal settlements of the Zeekoe catchment area. The specific objectives were: (i) To determine the risk factors for diarrhoea in informal settlements; (ii) To identify the distribution of diarrhoea in children under five years in this setting; and (iii) To recommend prevention and control measures for children’s diarrhoea in the low-lying informal settlements in Cape Town.

## 2. Materials and Methods

### 2.1. Ethics Statement

The study protocol was approved by the “Ethikkommission Nordwest—und Zentralschweiz (EKNZ)” in Switzerland (EKNZ BASEC 2016-00304) and by the “Human Research Ethics Committee (HREC)” of the University of Cape Town in South Africa (HREC REF 248/2016). All the interviews were conducted after the head of the household agreed to participate in the survey and informed consent forms were signed.

### 2.2. Study Location

The study was conducted in six informal and two formal settlements of Cape Town, located in the Zeekoe Catchment in the Cape Flats (a low-lying area of Cape Town). The map of the catchment study area is represented in Figure 1a–c. The catchment area overlays the Cape Flats Aquifer (CFA), a potential groundwater resource, and is marked by various land uses, precarious socio-economic conditions, black and grey water flows from settlements and industries, and a wastewater treatment plant (WWTP) that discharges treated effluent to the sea (Figure 1).

### 2.3. Selection of Settlements and Study Population

The Zeekoe Catchment area consists of a wide range of communities, with many formal settlements blended into a poor informal settlement and sharing the same polluted premises [19]. Six informal settlements in the catchment selected in a previous phase of the study were also selected for this study, namely: Barcelona, Lotus Park, Phola Park, Sweet Home, Weltevrede, and Pelican Park (Figure 1c). In addition, we selected nearby formal houses from Gugulethu and Manenberg. The selection of these sites was based on the following criteria established with the civil society actors working in the areas: (i) Accessibility of the area; (ii) Interviewer’s availability for the survey in the settlement; and (iii) Security issues. Of note, all the settlements from the catchment considered for inclusion in the study were high-risk crime areas, and thus, it was important to minimize the threat to the safety of the field staff. Areas were visited if site facilitators were available to accompany field staff. Houses in the selected areas were selected opportunistically.

### 2.4. Study Population

The sample size was chosen for use in a Monte Carlo simulation. A sample size of approximately 90 households per each settlement was selected, assuming that the assumed prevalence of childhood diarrhoea in study sites was 40% with a standard deviation of 5%, and the ratio between exposed and non-exposed groups was 1:1. ‘Diarrhoea’ was defined as “having three or more loose or liquid bowel movements over a 24-h period, as reported by the mother or caregiver of the child, at any point during the seven days preceding the interview” [7].

### 2.5. Data Collection

Data were collected from March to June 2017. The questionnaire was developed in English and translated into Xhosa and Afrikaans for appropriate communication within the local community, given that many residents are not fluent in the English language. A pilot study was carried out three days before the beginning of the household survey to identify possible shortcomings of the questionnaire and to verify the anticipated duration of the interview. The questionnaire covered the following sections: (i) basic demographic information, (ii) socio-economic status, (iii) water source, (iv) sanitation and hygiene, and (v) child health status. The survey questions were addressed to the head of the household or its legal representative at the time of the interview. The questionnaires were administered by six groups of trained interviewers consisting of two persons per group. The anonymity of the respondents was preserved throughout the course of the research.

### 2.6. Statistical Analyses

Statistical analyses were carried out using Stata14 software (StataCorp LP, Lakeway, TX, USA). Using factor analysis, a wealth index was created based on possessions of household items, and a hand-washing index was created based on hand-washing. All variables were screened for irregularities and outliers. Basic descriptive statistical analyses, such as median, mean, interquartile ranges, standard deviation (SD), and confidence interval (CI), were obtained for the relevant continuous variables, and a frequency table was generated for other variables.

Due to the hierarchical structure of collected data where child’s data being nested in the household and household data in communities, multi-level logistic modelling was performed [20]. Mixed-effects logistic regression modelling was used to examine the effects of child-related variables (level 1) and household-related variables (level 2) on childhood diarrhoea.

## 3. Results

### 3.1. Characteristics of Studied Households

#### 3.1.1. Demographic and Socio-Economic

A total of 707 households participated in the survey conducted in six informal settlements and two nearby formal settlements. There was a total of 778 children aged 0–5 years old in the recruited households. The demographic and health-related information of those children stratified by living location is shown in Table 1. The mean age of the children was 28.8 months (SD = 17), and 51.9% were female. While the gender of children was distributed equally in the IS, more females than males participated in FS (58.2% vs. 41.8%, respectively). The vast majority of caretaker respondents (>90%) were female. The median number of people per household in FS was 5 (SD: 2.2, range 1–18). It was also observed that residents of FS stay longer in their current address relative to those from the IS with an average duration of (23 years vs. 8 years, *p* = 0.005). Although FS families had a statistically significantly higher mean wealth index than those living in IS (*p* = 0.05), there was not much difference in the proportion of children going to daycare (47.1% vs. 47.3%, *p* = 0.99).

#### 3.1.2. Water Access and Resources

The majority of children in participating households had access to potable water (>97%) for domestic use. In IS, 74.4% of households used public taps as the main source of drinking water (Figure 2a). All residences of IS had access to water predominantly from a public tap, with the exception of Pelican Park that had piped tap water inside the yard as the main water resource (Figure 2b). In contrast, piped tap-water on the premises (89.6%) was the most common water resource in formal areas (Figure 2c).

Most IS residents reported storing water either in plastic bottles or containers (96.3%), with the water storage duration exceeding 12 h in 13.1% of households. FS residents were more likely to store water <12 h than those in IS (91.3% vs. 76.2%, Chi-square χ^2^ =26.9, *p* < 0.001). Only a small proportion (0.6%) of the caretakers, all from IS, reported that they need over 30 min to get to the water source. A general overview of access to water resources of participating households during the study is represented in Table 2.

#### 3.1.3. Sanitation

Most households (98.3%) reported access to latrines, but only 150 households (21.2%) had access to a flush toilet system. The proportion of IS households having access to an improved sanitation facility (modern toilet, potable water, waste disposal) was considerably lower than those of FS households (5.0% vs. 75.0%, Chi-square test, *p* < 0.001). Shared facilities were the most common in all IS (93%), while a private toilet was the most common among FS households (75%) (Figure 3a). The mean number of households that shared a sanitary facility in most informal households was 4 (range 1–11). Most of the sanitary facilities were located less than 50 m from the houses (94.8%). While flush toilets were the most frequently reported facility in formal areas (>96%) (Figure 3c), facility types varied across informal sites, with the flush toilet the most common in Pelican Park (96.6%) and Phola Park (97.7%); the portable toilet system is most common in Barcelona (71%) and Shengu (chemical toilets) and flush toilets in Lotus Park, Weltevrede, and Sweet Home (Figure 3b).

#### 3.1.4. Hygiene

With regard to hygiene, the participants were asked, “When do you usually wash your hands?” with options: before eating, after eating, after defecation, after latrine use, before feeding child, after handing rubbish, after handling child’s diaper/feces, before food preparation, after touching animals. The questionnaire also included the hygiene question, “Have you seen any rats/cockroaches around your house in the last 7 days.” Additionally, interviewers recorded the availability of soap in the house using an observation checklist.

The respondents reported whether they washed hands before eating (69.9% IS vs. 95.7% FS), after eating (61.1% IS vs. 55.5% FS), after defecation (51.2% IS vs. 37.2% FS), after latrine use (62.6% IS vs. 59.8% FS), before feeding child (61.9% IS vs. 61% FS), after handling rubbish (58% IS vs. 42.7% FS), after handling child’s diaper/feces (52.9% IS vs. 34.2% FS), before food preparation (58.9% IS vs. 64.6% FS), and after touching animals (12.5% IS vs. 12.2% FS). The results of an analysis using a hand-washing index derived from the above indicators using principal component analysis showed that respondents from IS had better hand-washing practices than FS (*t*-test, *p* = 0.02). The availability of soap at hand-washing places reported in both formal and informal settlements are similar (28.5% vs. 26.2%) and not statistically significantly different. With regard to wastewater disposal, most IS respondents reported that wastewater was disposed of onto open ground (63.8%), while most FS respondents reported that they disposed wastewater into the toilet (51.2%) and onto open ground far away from their houses (18.3%). Pest (rats, cockroaches) invasion 2 weeks prior to the survey occurred more frequently in IS compared to FS (87.0% vs. 46.0%, *p* = 0.00); however, using pesticides inside the house during 2 weeks prior to the survey was less frequently reported in IS than FS households (44.0% vs. 57.0%, *p* = 0.004). Hygiene practices of respondents from informal and formal households are presented in Table 3.

#### 3.1.5. Health-Related Indicators of Children

The health-related indicators for participating children include vaccines monitored by the South Africa Expanded Programme on Immunisation (EPI-SA). Children from FS were more frequently reported to be vaccinated than IS children: 74.1% vs. 50.9% for Hep B vaccination and 76.2% vs. 64.2% for Rotavirus, respectively (Chi-square test, *p* < 0.001). Interestingly, for HepA vaccination, which is not sponsored under EPI-SA, children living in informal households were reported to get injected more frequently than those living in formal households (45.0% vs. 33.9%, Chi-square test, *p* < 0.001). Overall, 12.3% of all children were breastfed exclusively for 6 months after birth. More than two-thirds of studied children (76.4%) reportedly had no breastfeeding at all. There was no statistically significant difference in breastfeeding rates between FS and IS children (21.1% vs. 24.5%, Chi-square test, *p* = 0.33).

### 3.2. Distribution of Diarrhoea Cases

The prevalence of diarrhoea among children under five years old was calculated based on the total number of cases that reported having diarrhoea during the 7 days prior to the interview in all the participating households. The distributions of diarrhoea cases by study site, age and gender distribution are illustrated in Figure 4.

Among a total of 778 recruited children, diarrhoea occurrence was recorded in 119 children (15.3%). The period prevalence was highest in Gugulethu (24.2%), followed by Pelican Park (19.4%), Manenberg (17.8%), Sweet Home (15%), Barcelona (14.4%), Lotus Park (11.1%), Phola Park (11.1%), and Weltevrede (8.6%) (Figure 4b). The prevalence of diarrhoea cases in this study was higher in formal settlements compared to informal settings (21.2% vs. 13.4% respectively, Chi-square test, χ^2^ = 6.6, *p* = 0.01) (Figure 4a). As seen in Figure 4c, the age group of 6–11 months had the highest prevalence of diarrhoea (26%), following by the group of 24–35 months (18%); conversely, the diarrhoea prevalence in other age groups was lower, ranging from 12%–13%. Diarrhoea prevalence was higher among girls than boys in all age groups, except in children aged 12–23 months. However, the gender difference was not statistically significant (Chi-square test, χ^2^ = 1.5, *p* = 0.22) (Figure 4d).

### 3.3. Risk Factors Associated with Diarrhoea Using Multi-Level Multivariate Analysis

The results from the multivariate analysis show that the caregiver’s education level was positively associated with the diarrhoea occurrence in children under five years old (OR =1.59, 95% CI 1.06–2.40, *p* = 0.03). There was no significant association between economic status, based on the wealth index and under-five-year child diarrhoea (OR = 0.67, 95% CI 0.38–1.17, *p* = 0.16). Usage of water stored >12 h showed a statistically significant positive association with diarrhoea (OR = 1.90, 95%CI 1.02–3.79, *p* = 0.05). Children from households who treat water before drinking were less likely to experience diarrhoea compared to children from households who do not treat water (OR = 0.57, 95%CI 0.34–0.97, *p* = 0.04). None of the drinking water sources were statistically significantly associated with diarrhoea. Interruption of water supply was not found to be statistically significantly associated (OR = 1.21, 95%CI 0.69–2.09, *p* = 0.5) with diarrhoea. Additionally, sanitation characteristics and ownership status, when summarized as a continuous variable, showed a significant association with under-five diarrhoea (OR = 0.47. 95% CI 0.26–0.84, *p* = 0.12). Interestingly, a higher number of households sharing the same facility was associated with a lower risk of diarrhoea among study participants. There was a significantly lower risk of diarrhoea among households sharing the facility with more than three other households compared to those using non-shared facilities (OR = 0.35, 95%CI 0.15–0.85, *p* =0.02). Dysfunctional toilet facilities showed a significant positive association with childhood diarrhoea (OR = 1.88, 95%CI 1.00–3.55, *p* = 0.05). An increased hand-washing index was found to be strongly associated with a lower risk of diarrhoea (OR = 0.61, 95%CI 0.47–0.79, *p* < 0.001). There was no significant association between under-five diarrhoea, gender, and age. The duration of breastfeeding showed a weak and statistically non-significant positive association with the risk of diarrhoea (OR = 1.11, 95%CI 1.00–1.23, *p* = 0.07). The results in Table 4 shows Hepatitis A vaccination to be a statistically significant protective factor (OR = 0.51, 95%CI 0.28–0.90, *p* = 0.02) while rotavirus vaccination was not statistically significantly associated with under-five diarrhoea (OR = 1.62, 95%CI 0.86–3.08, *p* = 0.14).

## 4. Discussions

This study revealed that socio-economic factors were important predictors of under-five diarrhoea. Children living in the formal settlements showed greater evidence of access to socio-economic facilities (accommodation, education and health) relative to those living in the informal settlements. It is important to note, however, that living in a formal settlement does not necessarily suggest better hygiene practice. People living in formal settlements have a surprisingly lower index of hand-washing compared to people living in informal settlements. The combination of poor living conditions, poor hygiene practices and contaminated environment in the neighbourhood contributed to diarrhoea cases in children under five in both formal and informal households. The literature suggests that the caregiver’s increasing level of education has been considered as a protective factor against diarrhoea [21,22,23]. However, the results in this study show that the higher education levels of caregivers are associated with a consistently increased risk of diarrhoea in children under five. The paradox has been found in some other studies reporting that there were no differences in knowledge of adequate dietary practices between different maternal education levels [24,25]. In addition, higher education could lead to earlier weaning which can increase episodes of diarrhoea in children [26].

Water access and resources is a crucial factor that serves as an indicator of community health. The majority of the investigated households in the formal settlements have access to the municipal water supply. Various factors relating to water access do not allow for any concrete conclusions about the safety of the system delivering water in informal households in this study. Inhabitants in informal settlements are not allowed to have private water connections to their homes; however, some make illegal connections without any supervision or instruction from local authorities, thus creating a path for potential pollution. There is seldom any system for draining the grey water generated by the in-house supply from the informal settlement [27].

Our findings show that sharing sanitation facilities with others was associated with an increased risk of diarrhoea relative to non-sharing facilities. Contamination of drinking water resources was higher in households with poor sanitation compared to those with adequate sanitation, and this may explain the increased risk of diarrhoea in the former. However, there are different types of shared sanitation services in informal settlements, varying from a self-maintained facility that is shared between a few nearby households to publicly run facilities (some might be maintained better than others). Although our data suggested that shared sanitation does not pose higher risks than private access, other factors related to sanitation could be more important in diarrhoea transmission. For instance, IS are constrained both with respect to space and water supply, and the majority of communal sanitation facilities are not linked to a sewage system. Stagnant water is often observed in such dense informal communities, which poses a big risk of spreading diarrhoea pathogens into the environment. Previous studies demonstrated children would face a high risk of diarrhoea from playing in soil and surface water outside the household [28,29]. Reinforcing the findings from other studies, we also found that dysfunctional toilet facilities to be significantly associated with childhood diarrhoea in this study. This study found that hygiene practices were associated with a reduced risk of childhood diarrhoea, which is consistent with the literature [30,31].

It is well-known that good hygiene could enhance healthy living, and treating point-use water reduces the risk of getting diarrhoea disease [30,32]. Our findings show that children from households with good hygiene are less likely to experience diarrhoea than those who did not. This study is consistent with similar studies elsewhere [33]. Hygiene practices, such as water purification methods, including water-boiling and filtration, could reduce the pathogen load in water if the process is carried out consistently and correctly according to standard techniques [30,34]. Another study also suggested that not all water treatment methods are effective if carried out in an unhygienic manner; improper infiltration could cause by unsafe storage water leading to recontamination after treating [35]. Our data shows that drinking water stored longer than 12 h could contribute to the potential risk of childhood diarrhoea. Similar results were found in other studies conducted in developing countries [36,37,38]. Our findings suggested that sharing a facility with more than four households significantly lower the risk of diarrhoea compared to those using a non-shared facility. Similar studies conducted in Tanzania showed that shared latrines contained lower concentrations of Escherichia coli compared to private latrines [39]. Both results are unexpected. Other studies and meta-analyses found that shared sanitation is generally dirtier and associated with an increased risk of diarrhoea compared to private sanitation [40,41,42,43], suggesting the need for further investigation.

### 4.1. Child-Health Related Factors

In this study, the highest diarrhoea prevalence occurred in children aged 06–11 months. This finding is consistent with that of previous research that suggests diarrhoea is more likely to happen to younger children under 12 months, and the risk reduces with increasing age [44,45,46]. Ordinarily, breastfeeding should increase the children’s immune system and thus protect them against diarrhoea, but this was not found to be the case for those aged under five in this study. This could be due to exposure to contaminated water lying around the informal settlements, increased exposure of the toddler to contaminated water and/or lack of personal hygiene. Additionally, practicing breastfeeding may contribute to childhood diarrhoea if the nursing mothers engage in unhygienic practices such as dirty breasts. Findings also revealed that children who took the HepA vaccine are at lower risk than those who were not HepA vaccinated. The prevalence of Hep A infection in the study is likely due to exposure to poor water and sanitation conditions. Serological studies detected HepA in fecal samples of children with acute diarrhoea in developing countries [47,48,49]. Furthermore, a number of studies have reported the detection or isolation of Hep A virus from water resources, with the detection rate ranging from 76% in surface water to 37% in wastewater plants and irrigation water [50,51]. However, there was no available data on the Hep A prevalence in children under five years in South Africa, and the role of Hep A virus as an element in diarrhoea has not been documented. The results from this study stress the need for further investigation with routine tracking of Hep A infection as a contributory cause of childhood diarrhoea in sub-urban areas in Cape Town.

### 4.2. Strengths and Limitations

The strengths of this study include the opportunity to estimate the effects of multiple environmental factors and diarrhoea burden using individual-level data from a community-based survey and using multi-level analyses to consider the effects of individual-level and household-level factors in one analysis. Among the limitations of this study, we did not explore the comparison of the ways sanitation facilities are shared in informal settlements with private access facilities. Most households in informal settlements had access to only shared sanitation facilities. There are limited data on how shared sanitation facilities in informal settlements were managed and maintained. In addition, documentation of how stagnant water and leakages were contained and managed was not explored by our study. Those factors may be important sanitation-related confounders on the effects of sanitation on childhood diarrhoea. We were unable to check our data on diarrhoea prevalence against that collected by the local health authorities owing to the difficulty in obtaining the necessary approvals within the limited time frame of the study. A standard limitation is that diarrhoea was reported by respondents with no clinical confirmation. This method is widely used, although clinical confirmations make the disease classification more reliable. The study also did not track the casual, social, and permanent movement of people from informal settlements to nearby formal houses with exposure to different types of water sources and sanitation facilities, which might contribute to the risk of getting diarrhoea. Moreover, the cross-sectional study design only allows us to describe associations but does not allow us to infer causality. Finally, the study did not involve any biomedical tests which could provide stronger information on diarrhoea aetiology.

## 5. Conclusions

The study found several factors that were significantly associated with the increased risk of diarrhoea among children under five years old living in informal and formal settlements of the Lotus River catchment area in Cape Town. Namely, these factors are caregiver’s education level, storing water longer than 12 h, not using proper water treatment, sanitation facility shared with more than four households, dysfunctional toilet facility, using pesticides inside the houses, poor hand-washing practices, longer duration of breastfeeding, and not having Hepatitis A vaccination. It was clear that good hygiene practices are key in the prevention of diarrhoea. Based on those findings, the promotion of hygienic practices is highly recommended. Caregivers should have more knowledge about and exercise better the practices of hand-washing, hygienic breastfeeding, and food preparation. Proper and healthy water storage is necessary, covering water containers, and basic water treatment techniques, such as boiling, filtration, and disinfection, should be observed. Local authorities should support improvement on the adequate sanitation and drainage service provision in both formal and informal settlements. A periodic water quality monitoring campaign at the communal level will be necessary to identify early enough possible water contamination problems, which can lead to diarrhoea. Finally, further studies on diarrhoea aetiology and transmission pathways in the informal urban context are required to get a better understanding of diarrhoea causality in these communities.

## Figures and Tables

**Figure 1 ijerph-18-06043-f001:**
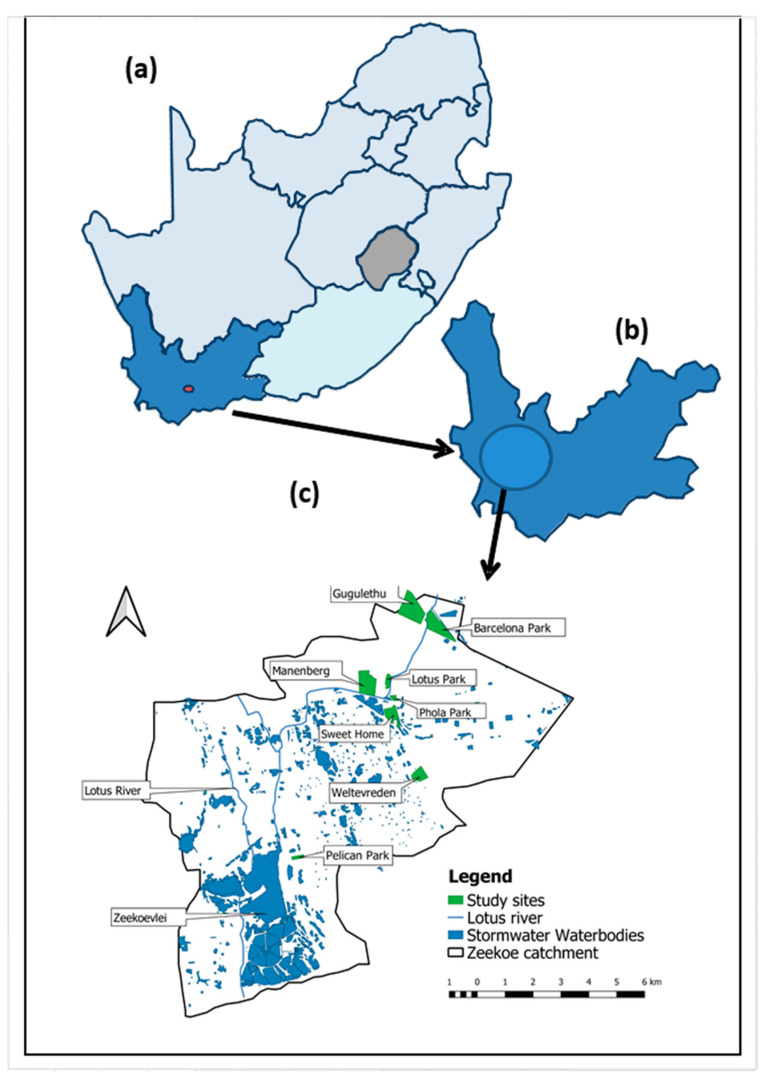
Map of South Africa (**a**), Western Cape (**b**), and the catchment area in Cape Town showing the six informal settlements in the (Zeekoe catchment) and different land use (**c**).

**Figure 2 ijerph-18-06043-f002:**
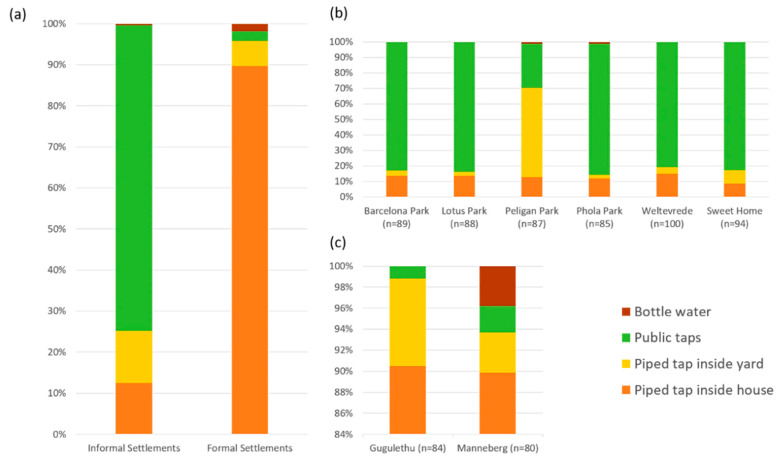
Types of drinking water sources (**a**) in each settlement type; (**b**) in six informal settlements (**c**) in two formal settlements.

**Figure 3 ijerph-18-06043-f003:**
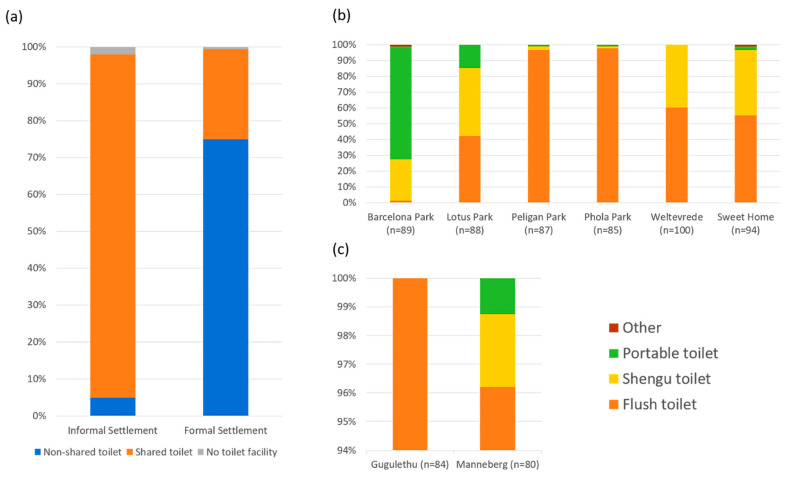
Types of sanitation facilities available (**a**) in each settlement type; (**b**) in six informal settlements; and (**c**) in two formal settlements.

**Figure 4 ijerph-18-06043-f004:**
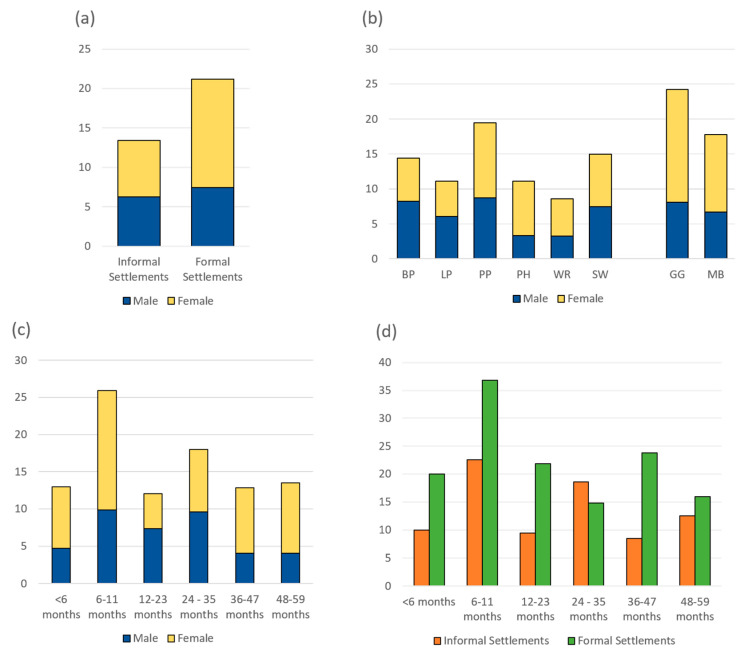
(**a**) The prevalence of diarrhoea cases by gender in each settlement type; (**b**) The prevalence of diarrhoea cases by gender in each study site; (**c**) The distribution of diarrhoea cases by gender in each age group; and (**d**) The prevalence of diarrhoea cases by house types in each age group. Note: BP (Barcelona Park), LP (Lotus Park), PP (Peligan Park), PH (Phola Park), WR (Weltevrede), SH (Sweet Home), GG (Gugulethu), and MB (Manenberg).

**Table 1 ijerph-18-06043-t001:** Demographic and health-related characteristics of children in participating households.

Sites	Overall (N = 778)	Informal Settlements (IS)							Formal Settlements (FS)		
		Overall IS (n = 589)	Barcelona (n = 97)	Lotus Park (n = 99)	Pelican Park (n = 103)	Phola Park (n = 90)	Weltevrede (n = 93)	Sweet Home (n = 107)	Overall FS (n = 189)	Gugulethu (n = 99)	Manenberg (n = 90)
**Gender**											
Male	374 (48.1%)	295 (50.1%)	54 (55.7%)	54 (54.6%)	50 (48.5%)	41 (45.6%)	47 (50.5%)	49 (45.8%)	79 (41.8%)	40 (40.4%)	39 (43.3%)
Female	404 (51.9%)	294 (49.9%)	43 (44.3%)	45 (45.4%)	53 (51.5%)	49 (54.4%)	46 (49.5%)	58 (54.2%)	110 (58.2%)	59 (59.6%)	51 (56.7%)
**Age group (6 months intervals)**											
<6	85 (10.9%)	60 (10.2%)	6 (6.2%)	7 (7.1%)	10 (9.7%)	5 (5.6%)	18 (19.4%)	14 (13.1%)	25 (13.2%)	15 (15.2%)	10 (11.1%)
6–11	81 (10.4%)	62 (10.5%)	9 (9.3%)	8 (8.1%)	14 (13.6%)	6 (6.7%)	9 (9.7%)	16 (15%)	19 (10.1%)	13 (13.1%)	6 (6.7%)
12–23	149 (19.2%)	117 (19.8%)	12 (12.4%)	25 (25.3%)	18 (17.5%)	23 (25.6%)	17 (18.3%)	22 (20.6%)	32 (16.9%)	17 (17.2%)	15 (16.7%)
24–35	167 (21.5%)	140 (23.8%)	23 (23.7%)	26 (26.3%)	32 (31.1%)	22 (24.4%)	16 (17.2%)	21 (19.6%)	27 (14.3%)	12 (12.1%)	15 (16.7%)
36–47	148 (19%)	106 (18%)	26 (26.8%)	17 (17.2%)	18 (17.5%)	12 (13.3%)	18 (19.4%)	15 (14%)	42 (22.2%)	22 (22.2%)	20 (22.2%)
48–59	148 (19%)	104 (17.7%)	21 (21.7%)	16 (16.2%)	11 (10.7%)	22 (24.4%)	15 (16.1%)	19 (17.8%)	44 (23.3%)	20 (20.2%)	24 (26.7%)
**Vaccination for child**											
HepA	329 (42.3%)	265 (45%)	37 (38.1%)	57 (57.6%)	55 (53.4%)	45 (50%)	28 (30.1%)	43 (40.2%)	64 (33.9%)	45 (45.5%)	19 (21.1%)
Hep B	522 (67.1%)	378 (64.2%)	53 (54.6%)	59 (59.6%)	93 (90.3%)	70 (77.8%)	43 (46.2%)	60 (56.1%)	144 (76.2%)	69 (67.7%)	75 (83.3%)
Rotavirus	440 (56.6%)	300 (50.9%)	59 (60.8%)	41 (41.4%)	62 (60.2%)	54 (60%)	45 (48.4%)	39 (36.5%)	140 (74.1%)	62 (62.6%)	78 (86.7%)
**Breastfeeding in the first 06 months**											
No breastfeeding	594 (76.4%)	445 (75.6%)	77 (79.4%)	76 (76.8%)	62 (60.2%)	76 (84.4%)	72 (77.4%)	82 (76.6%)	149 (78.8%)	82 (82.8%)	67 (74.4%)
Partial	88 (11.3%)	73 (12.4%)	11 (11.3%)	13 (13.1%)	23 (22.3%)	8 (8.9%)	8 (8.6%)	10 (9.4%)	15 (7.9%)	9 (9.1%)	6 (6.7%)
Inclusive	96 (12.3%)	71 (12.1%)	9 (9.3%)	10 (10.1%)	18 (17.5%)	6 (6.7%)	13 (14%)	15 (14%)	25 (13.2%)	8 (8.1%)	17 (18.9%)
**Having diarrhea during 07 days prior to the survey**											
Yes	119 (15.3%)	79 (15.3%)	14 (14.4%)	11 (11.1%)	20 (19.4%)	10 (11.1%)	8 (8.6%)	16 (15%)	40 (21.2%)	24 (24.2%)	16 (17.8%)

n = Number of children of participating households.

**Table 2 ijerph-18-06043-t002:** Access to water resources of participating households from informal and formal settlements.

Sites	Overall N = 707	Informal Settlements (IS)							Formal Settlements (FS)		
		Overall IS (n = 543)	Barcelona (n = 89)	Lotus Park (n = 88)	Pelican Park (n = 87)	Phola Park (n = 85)	Weltevrede (n = 100)	Sweet Home (n = 94)	Overall FS (n = 164)	Gugulethu (n = 84)	Manenberg (n = 80)
**Access to water resources**	99.4%	99.3%	97.8%	100%	100%	100%	98%	100%	100%	100%	100%
**Drinking water resources**											
Piped tap inside house	30.4%	12.5%	13.5%	13.6%	12.6%	11.8%	15%	8.5%	89.6%	89.4%	89.9%
Piped tap inside yard	11.2%	12.7%	3.4%	2.3%	57.5%	2.4%	4%	8.5%	6.1%	8.3%	3.8%
Public taps	57.7%	74.4%	83.2%	84.1%	28.7%	84.7%	81%	83%	2.4%	2.4%	2.5%
Use of bottled water	0.7%	0.4%	0	0	1.2%	1.2%	0	0	1.8%	0	3.8%
**Distance to water resource**											
Within 500 m	95.5%	94.1%	89.9%	96.6%	100%	100%	92%	87.2%	100%	100%	100%
>500 m	4.5%	5.9%	10.1%	3.4%	0	0	8%	12.8%	0	0	0
**Time spent to fletch water**											
Within 30 min	99.4%	99.3%	97.8%	100%	100%	100%	98%	100%	100%	100%	100%
>30 min	0.6%	0.7%	2.2%	0	0	0	2%	0	0	0	0
**Storing water <12 h**	87.8%	91.3%	88.8%	100%	86.2%	89.4%	93%	90.4%	76.2%	87.1%	64.6%
**Water cut-off in the last 30 days**	19.4%	20.1%	36%	16%	6.9%	5.9%	42%	10.6%	17.1%	10.6%	24.1%
**Treating water before drinking**	46.5%	42.7%	28.1%	53.4%	57.5%	36.5%	50%	30.9%	59.2%	45.9%	73.2%
**Type of water-storing container**											
Plastic bucket/bottles	96.3	98.3%	100%	98.9%	98.8%	97.6%	97%	98.9%	89.6%	85.7%	93.8%
Other	3.7%	1.7%	0	1.1%	1.2%	2.4%	3%	1.1%	10.4%	14.3%	4.2%
**Cleaning frequency of water-storing container**											
Daily	79.5%	82%	69.7%	90.9%	90.8%	789.4%	75%	77.7%	71.3%	77.7%	64.6%
<1 time/day	120.5%	18%	40.3%	9.1%	9.2%	10.6%	25%	22.3%	29.7%	22.3%	35.4%

**Table 3 ijerph-18-06043-t003:** The hygiene in participating households from informal and formal settlements.

	Overall (N = 707)	Informal Settlements (IS)							Formal Settlements (FS)		
		Overall IS (n = 543)	Barcelona (n = 89)	Lotus Park (n = 88)	Pelican Park (n = 87)	Phola Park (n = 85)	Weltevrede (n = 100)	Sweet Home (n = 94)	Overall FS (n = 164)	Gugulethu (n = 84)	Manenberg (n = 80)
**Respondent’s hand-washing practices**											
Before eating	96.6%	96.9%	97.8%	90.9%	100%	100%	96%	96.8%	95.7%	94.1%	97.5%
After eating	59.8%	61.1%	57.3%	53.4%	82.8%	77.7%	51%	47.9%	55.5%	63.5%	46.8%
After defecation	48%	51.2%	39.3%	53.4%	66.7%	61.2%	42%	46.8%	37.2%	50.6%	22.8%
After latrine use	62%	62.6%	58.4%	59.1%	83.9%	72.9%	50%	54.3%	59.8%	64.7%	54.4%
Before feeding child	61.7%	61.9%	55.1%	56.8%	80.5%	53.2%	51%	77.7%	61%	63.5%	58.2%
After handling rubbish	54.5%	58%	50.6%	55.7%	67.8%	75.3%	51%	50%	42.7%	58.8%	25.3%
After handling baby diaper/feces	48.5%	52.9%	41.6%	50%	69%	68.2%	46%	44.7%	34.2%	52.9%	13.9%
Before food reparation	60.3%	58.9%	49.4%	59.8%	87.4%	64.7%	48%	50%	64.6%	57.7%	72.2%
After touching animals	12.5%	12.5%	9%	8%	23%	24.7%	3%	9.6%	12.2%	18.8%	5%
Observed availability of soap	28%	28.5%	31.5%	51.1%	24.1%	15.3%	15%	35.1%	26.2%	42.1%	12.5
Hand-washing index (mean)	−2.30 × 10^−8^	0.04	−0.14	−0.05	0.5	0.4	−0.16	−0.2	−0.14	0.06	−0.36
**Pest invasion**											
Rat	77.4%	86.9%	92.1%	94.3%	61%	80%	94%	97.9%	45.7%	72.9%	16.5%
Cockroaches	70.9%	71.8%	78.7%	83%	60.9%	57.7%	81%	68.1%	67.7%	58.8%	77.2%
**Using pesticides during the last 2 weeks**											
Inside house	47.4%	44.4%	53.4%	26.4%	50.6%	41.2%	54.6%	39.4%	57.3%	37.7%	78.5%
Outside house	21.6%	21.7%	22.5%	18.2%	25.3%	12.9%	26.3%	23.9%	21.3%	28.3%	13.9%
**Waste water disposal**											
To open ground near house	12.2%	10.9%	14.6%	3.4%	4.6%	33.5%	17%	20.2%	16.5%	2.4%	31.7%
To open ground far away from house	44.8%	52.9%	58.4%	52.3%	18.4%	69.4%	61%	56.4%	18.3%	31.8%	3.8%
To toilet/ lavabo	33.7%	28.4%	22.5%	42.1%	62.1%	17.7%	9%	20.2%	51.2%	57.7%	44.3%
To water storm drains	9.3%	7.9%	4.5%	2.3%	14.9%	9.4%	13%	3.2%	14%	8.2%	20.3%

**Table 4 ijerph-18-06043-t004:** The multi-level multivariate mixed-effects logistic regression analysis of risk factors associated with diarrhoea.

Risk Factors	OR	95% CI	*p*-Value
Caretaker’s education level	1.59	1.06–2.40	0.03
Having >2 U5-children	0.67	0.38–1.17	0.16
Drinking water sources			
Private piped taps on premises	Ref	-	-
Piped taps inside yard	1.94	0.84–4.47	0.12
Communal public tap	0.59	0.14–2.54	0.48
Storing water >12 h	1.9	1.02–3.79	0.05
Using water treatment	0.57	0.34–0.97	0.04
Water interruption during 07 days prior the survey	1.21	0.69–2.09	0.50
Toilet access and sharing			
Private toilet facility	Ref	-	-
Shared with 1–3 households	0.76	0.34–1.64	0.49
Shared with ≥4 households	0.35	0.15–0.85	0.02
No toilet facility	0.44	0.07–2.77	0.39
Reported problems with toilet facility	1.88	1.00–3.55	0.05
Feces disposal	1.18	0.75–1.84	0.47
Using pesticide inside house during 07 days prior the survey	0.49	0.30–0.84	0.01
Hand-washing index	0.59	0.42–0.82	0.02
Child’s age (in months)	0.99	0.98–1.01	0.23
Child’s gender			
Male	Ref	-	-
Female	1.32	0.83–2.09	0.23
Duration of breast feeding	1.11	1.00–1.23	0.07
Had Hepatitis A vaccine	0.51	0.28–0.9	0.02
Had Rotavirus vaccine	1.62	0.86–3.08	0.14

N.B: Results of multi-level multivariate analyses of risk factors associated with diarrhoea are presented, including the following groups of variables: socio-demographic (caretaker education level, having other under-5 children in the household and wealth index), water-related variables (drinking water sources, water storage >12 h, water treatment options and water interruption during within the period of seven 7 days prior the survey), sanitation and hygiene-related variables (facility access and sharing, reported a problem with toilet facility, disposal of child’s feces, hand-washing index), and children-specific variables (child’s age, gender, breastfeeding duration and immunization history of Hepatitis A and Rotavirus vaccine).

## Data Availability

The data presented in this study are available from the corresponding author on reasonable request. The data are not publicly available due to the presence of confidential participants’ information.

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
