# Peer review of "Diarrhoea among Children Aged under Five Years and Risk Factors in Informal Settlements: A Cross-Sectional Study in Cape Town, South Africa"

_ijerph, 2021, doi:10.3390/ijerph18116043_

Round 1
Reviewer 1 Report
Dear Authors,
before publication, please consider the following comments:
2.3. I understand that the selection of units was deliberate and not random? Please describe exactly what safety considerations were taken into account?
3.1.4. Hygiene - please explain what exactly was the question in the survey? Do the results mean that 69.9% of IS always wash their hands before eating?
Here the results may depend on the form of the question asked; "Do you generally wash your hands?" "Do you ever fail to wash your hands?" Or: "Did you wash your hands before the last meal you eat?"
The restriction related to the testing tool of the questionnaire should be added to the appropriate section.
Besides, a very interesting job. Thank you.
Author Response
- I understand that the selection of units was deliberate and not random? Please describe exactly what safety considerations were taken into account?
Response: We added the following text to the end of section 2.3 to clarify the safter considerations and household selection.
Line 119 – 122: “Of note, all the settlements from the catchment considered for inclusion in the study were high-risk crime areas and thus it was important to minimize the threat to the safety of the field staff. Areas were visited if site facilitators were available to accompany field staff. Houses in the selected areas were selected opportunistically”
- In the Data Collection section (Section 2.5), it is indicated that houses were selected randomly.
Hygiene - please explain what exactly was the question in the survey? Do the results mean that 69.9% of IS always wash their hands before eating?
Here the results may depend on the form of the question asked; "Do you generally wash your hands?" "Do you ever fail to wash your hands?" Or: "Did you wash your hands before the last meal you eat?"
Response:
- To clarify the reviewer’s question of “Hygiene - please explain what exactly was the question in the survey?”, we added the following text at the start of section 3.1.4:
Line 212 – 218: “With regard to hygiene, the participants were asked “When do you usually wash your hands?” with options: before eating, after eating, after defecation, after latrine use, before feeding child, after handing rubbish, after handling child’s diaper/feces, before food preparation, after touching animals. The questionnaire also included the hygiene question “Have you seen any rats/cockroaches around your house in the last 7 days. Additionally, interviewers recorded the availability of soap in the house using an observation checklist.”
All the above variables were included to construct a hand-washing index using principal factors analysis as stated in line 224-225: “Results of an analysis using a hand-washing index derived from the above indicators using principal component analysis showed….”
- To answer the reviewer’s question of “Do the results mean that 69.9% of IS always wash their hands before eating?”
According the study results, 69.9% of participants reported to wash their hands before eating in general. However, as it’s self-reported, we can’t assure that people are always washing their hand before eating in the reality.
Reviewer 2 Report
The paper entitled “Diarrhoea among children aged under five years, risk factors and hygiene practices in informal settlements of the Zeekoe catchment in Cape Town, South Africa: results from a cross-sectional study”
aimed to investigate diarrhoea among children aged under five years and the associated risk factors in informal settlements of Cape Town, South Africa.
The manuscript is well written and well supported by consistent references. The conclusions are largely sound and improve the existing knowledge.
Some suggestions to improve the manuscript are reported below.
- The title is too long, it should be more concise.
- The abstract exceeds the word limit required by the journal (200 vs 322). Authors should, for example, avoid repeating the results in the conclusions paragraph.
- Page 2 line 48 and page 3 line 110: the brackets must be removed
- Page 4 in the caption of figure 1 the reference to letter c is missing
- Page 5 line 64 the authors write: "The demographic and health-related information of those children stratified by living location is shown in Table 1" this sentence is inconsistent with the caption "Age distribution of children in participating households" and with the data shown in table 1
- Line 185 the authors wrote (Fig 2a-c) while they should write Fig.2c
- Line 209 the authors wrote (Fig 3b-c) but did not really comment on fig. 3c.
- Lines 211-213: why do the authors report the following notes? Notes: GG (Gugulethu), MB 211 (Manenberg), BP (Barcelona Park), LP (Lotus Park), PP (Peligan Park), PH (Phola Park), WR (Weltevrede), SH (Sweet 212 Home). In the figure the full names are written so I suggest deleting the notes.
- In the "discussion" paragraph the authors often report the results (lines 348 - 365, 376, 381), the results should not be repeated in this paragraph.
Author Response
Some suggestions to improve the manuscript are reported below.
1. The title is too long, it should be more concise.
Response: We changed the title to “Diarrhoea among children aged under five years and risk factors in informal settlements: a cross-sectional study in Cape Town, South Africa”
2. The abstract exceeds the word limit required by the journal (200 vs 322). Authors should, for example, avoid repeating the results in the conclusions paragraph.
Response: We have modified and reduced the word count for the abstract to 200 words as required.
3. Page 2 line 48 and page 3 line 110: the brackets must be removed
Response: We have removed brackets in line 48 and line 110.
4. Page 4 in the caption of figure 1 the reference to letter c is missing
Response: We have added letter (c) in the reference of Figure 1.
5. Page 5 line 64 the authors write: "The demographic and health-related information of those children stratified by living location is shown in Table 1" this sentence is inconsistent with the caption "Age distribution of children in participating households" and with the data shown in table 1
Response: We have addressed this issue and added demographic data on children in Table 1.
6. Line 185 the authors wrote (Fig 2a-c) while they should write Fig.2c
Response: We have changed (Fig 2a-c) to (Fig 2c) as in line 181 of the revised version.
7. Line 209 the authors wrote (Fig 3b-c) but did not really comment on fig. 3c.
Response:
Comment on Fig 3c was written on lines 204-205 “While flush toilets were the most frequently reported facility in formal areas (>96%) (Fig 3c)…”
However, to make it clearer, we have added (Fig 3c) in line 205 and changed (Fig 3b-c) to (Fig 3b) in line 208 as the reviewer’s comment.
8. Lines 211-213: why do the authors report the following notes? Notes: GG (Gugulethu), MB 211 (Manenberg), BP (Barcelona Park), LP (Lotus Park), PP (Peligan Park), PH (Phola Park), WR (Weltevrede), SH (Sweet 212 Home). In the figure, the full names are written so I suggest deleting the notes.
Response: We have deleted the notes in Figure 3 title as in line 210 of the revised version.
9. In the "discussion" paragraph the authors often report the results (lines 348 - 365, 376, 381), the results should not be repeated in this paragraph.
Response: In the discussion part, we have deleted the results (reported OR, 95% CI) as suggested in line 352, line 358, line 366, line 369, line 380, and line 385.